# Range Extension of Borehole Strainmeters Using MOSFET-Based Multi-Switch Automatic Zero Setting

**DOI:** 10.3390/s25020476

**Published:** 2025-01-15

**Authors:** Chen Yang, Zheng Chen, Hong Li, Wenbo Wang, Weiwei Zhan, Liheng Wu, Yunkai Dong, Jiaxin Chen

**Affiliations:** 1National Institute of Natural Hazards, Beijing 100085, China; 2School of Emergency Management Science and Engineering, University of Chinese Academy of Sciences, Beijing 100049, China

**Keywords:** borehole strainmeter, range extension, automatic control, field-effect transistor

## Abstract

Borehole strainmeters are essential tools for observing crustal deformation. In long-term observational applications, the dynamic changes in crustal deformation over multi-year scales often exceed the single measurement range of borehole strainmeters. Expanding the measurement range while maintaining high precision is a critical technical challenge. To address this, a full-range measurement system was developed using a bidirectional analog multi-switch based on MOS transistors and automatic feedback control. This system automatically adjusts the zero point of the measurement bridge, maintaining the bridge output at a near-balanced state. The quantifiable zero-setting actions are dynamically converted into equivalent voltage, enabling automatic full-range measurements while fully utilizing the effective linear range of the differential capacitive sensors. A laboratory performance tests demonstrated that an RZB borehole strainmeter equipped with this automatic zero-setting range extension system successfully covers the differential capacitive sensor’s effective linear range of approximately 100 μm.

## 1. Instruction

Variations in crustal stress and strain fields are the fundamental drivers of geodynamic processes such as rock layer fractures and earthquakes [1]. Borehole strainmeters play a critical role in crustal deformation monitoring [2,3], with a high-frequency range comparable to that of seismographs [2,4] and a low-frequency range overlapping with the observational capabilities of GPS and InSAR. These attributes make borehole strainmeters ideal for capturing short-term continuous crustal deformation [5,6]. Since the 1980s, countries such as China, the United States, and Japan have implemented borehole strain observation programs [7,8,9,10,11]. As an early adopter, China has developed various borehole strainmeter models, including the RZB, SKZ, and YRY types. Over three decades of advancement, borehole strain observation has become an essential tool in crustal deformation studies [2,3,12], contributing significantly to geophysics, seismology, and geodesy [13].

The four-component borehole strain measurement system typically comprises a strain probe, a host unit, and a means of data acquisition. The strain probe is installed within the borehole, where displacement sensors monitor the deformation of the borehole aperture. The host unit contains the measurement circuits for the sensors and is connected to the downhole probe via cables. The data acquisition system collects and stores the data from sensor measurements, analyzing them to obtain the relative deformation state of the surrounding rock formations and its dynamic evolution [14]. The strain probe usually adopts a steel cylinder casing structure, as shown in Figure 1, and is coupled to the borehole bedrock using cement.

The micro-displacement sensors are evenly distributed inside the steel cylinder at 45° intervals, and fixed at both ends to the inner wall of the casing [15]. When subjected to uniform horizontal principal strains *ε*_1_ and *ε*_2_ at a distance, the theoretical relative change in the borehole aperture in the direction θ of a given sensor is as follows:(1)$$S_{\theta }=A\left(\varepsilon_{1}+\varepsilon_{2}\right)+B\left(\varepsilon_{1}-\varepsilon_{2}\right)\;\cos\left(\theta -\varphi \right)$$

Here, θ is the angle of the sensor arrangement, φ is the angle of the maximum principal strain, A and B are related to the effective Young’s modulus and Poisson’s ratio of the cylinder and surrounding rock, and the relative deformation $S_{\theta }$ is measured by the sensor. By solving the simultaneous equations from four sensors, the relative deformations in the uniform horizontal principal strains *ε*_1_ and *ε*_2_, as well as the direction φ, can be determined [16]. The sensors typically comprise a differential capacitive micro-displacement sensor and form a ratio measurement bridge with the ratio transformer in the host unit [17]. Crustal deformation causes displacement of the capacitor plates, generating an unbalanced signal in the bridge. This signal is processed via impedance transformation, AC amplification, phase-sensitive detection, and low-pass filtering, and converted into a voltage representing the displacement of the capacitor plate by an analog-to-digital converter (ADC). This voltage is subsequently used to calculate the strain [3,17].

Due to the need for long-term monitoring, borehole strainmeters must accommodate significant crustal deformation. Large, unbalanced signals from the measurement bridge may exceed the limits of downstream processing circuits, requiring the bridge to be rebalanced, a process known as zero setting [6]. Electrical zero setting is typically achieved by adjusting the transformer tap point to rebalance the bridge, while mechanical zero setting involves physically repositioning the capacitor’s central plate.

Mechanical adjustments allow the bridge to be rebalanced over a broader range, thus expanding the measurement range [18]. However, these adjustments introduce mechanical components that increase the system complexity and risk of failure. In contrast, electrical zero setting uses a multi-tap ratio transformer, which effectively acts as an equivalent transformer with 10,000 tap windings. When the bridge circuit output exceeds the limits of downstream processing, adjusting the transformer tap modifies the voltage division ratio, reestablishing balance.

Using the RZB strainmeter as an example, the measurable effective plate spacing of its differential capacitive sensor is approximately 300 µm. Without mechanical zero setting, the sensor’s total effective linear range in practice is roughly 100 µm, as shown in Figure 2. The displacement of the capacitor plates within this range is considered a single adjustment measurement span, typically about 1 µm, which is significantly smaller than the total sensor range [19]. If the zero setting is not promptly adjusted during measurements, issues such as data discontinuities and breakpoints can arise. Moreover, during periods of rapid crustal deformation, frequent manual adjustments increase the operational workload at observation stations.

In 1984, Gladwin et al. proposed an automatic zero-setting approach using relay switches to control the selection of transformer taps [11]. While relay-based mechanical switches enable lossless signal transmission, they suffer from limitations such as unreliable contacts, slow switching speeds, and low integration. In this study, we present a fully electronic automatic zero-setting measurement system utilizing MOSFET-based multi-switches. This system enables the full-range measurement of the sensor, effectively utilizing the linear range of the capacitive sensor [19]. Additionally, the electronic solution offers advantages including high reliability, high integration, low power consumption, and fast switching.

## 2. Automatic Zero-Setting Measurement System Design

This paper presents the design of a full-range measurement system for differential capacitive micro-displacement sensors based on the automatic zero setting. As depicted in Figure 3, the system consists of a differential capacitive micro-displacement sensor, a radiometric measurement bridge circuit, a microcontroller-driven MOSFET multi-stage switch array, and a data acquisition and automatic control system. The differential capacitive displacement sensor, combined with the ratio transformer, forms the measurement bridge. When the capacitive plates are displaced, the unbalanced signal generated by the bridge undergoes impedance transformation, amplification, phase-sensitive detection, and low-pass filtering before being captured by the ADC, producing a voltage output proportional to the displacement of the plates.

If the displacement of the capacitive plates exceeds the threshold, the control system automatically selects the appropriate MOS transistor switch to engage the corresponding tap on the ratio transformer. This action realigns the measurement bridge to a near-balanced state. The automatic zero setting operation is quantifiable, and the data acquisition system, in combination with the bridge’s actual output voltage and the quantified zero-setting voltage, calculates an equivalent voltage representing the displacement of the intermediate plates. This process achieves full-range measurement capabilities.

### 2.1. Principle of Range Extension

Range extension is achieved by dynamically zero-setting the measurement bridge, ensuring that the output voltage of the intermediate plate remains within the circuit and ADC limits. By quantifying the zero-setting process and integrating it with the actual output voltage of the bridge, the system calculates a wide range of equivalent voltage values. This approach removes the limitations imposed by the circuit and ADC range, thus extending the system’s overall measurement range. The measurement bridge comprises a differential capacitive sensor and an equivalent transformer with N turns, as shown in the equivalent circuit diagram in Figure 4.

The differential capacitive micro-displacement sensor includes three capacitive plates: two fixed plates (upper and lower) with a combined physical spacing of *d*_1_ + *d*_2_ = 0.5 mm and one movable intermediate plate, forming two differential capacitors. The transformer is modeled as having 10,000 turns, divided into two inductances at a variable tap point such that *N*_1_ + *N*_2_ = 10,000. Together with the differential capacitors, the transformer forms an AC Wheatstone measurement bridge. To improve the output signal’s signal-to-noise ratio and reduce parasitic capacitance interference, the excitation signal’s Us voltage should be maximized and the frequency minimized within feasible limits. In practice, based on long-term practical experience, *U_S_* is typically selected as a 50 V, 781 Hz AC signal.

The unbalanced signal ${\dot{U}}_{O}$ is output from the intermediate plate of the capacitive sensor to the ground. In the ideally balanced state, the ratio of the gaps between the upper and lower plates, *d*_1_*/d*_2_, equals the turns ratio of the inductances on either side of the tap point, *N*_1_*/N*_2_, which is also equal to the voltage division ratio of the bridge, ${\dot{U}}_{1}/{\dot{U}}_{2}$. Under these conditions, the bridge output voltage ${\dot{U}}_{O}$ = 0, and the relationship can be expressed as Equation (2):(2)$$\frac{d_{1}}{d_{2}}=\frac{N_{1}}{N_{2}}=\frac{{\dot{U}}_{1}}{{\dot{U}}_{2}}$$

When the intermediate plate is displaced by ∆d, it generates an unbalanced voltage signal ${\dot{U}}_{O}$, which is amplified, phase-sensitively detected, and filtered to obtain the effective voltage value. The ADC collects the final output voltage $U_{O}$, which is proportional to the displacement ∆d of the intermediate plate. If the displacement ∆d is large, the resulting ${\dot{U}}_{O}$ may exceed the backend circuit’s limits, necessitating a reconfiguration of the transformer’s tap connection. This adjustment modifies the turns ratio of the transformer to *(N*_1_
*− n)/(N*_2_
*+ n)*, restoring the bridge to a near-balanced state.

This zero-setting process is quantifiable, with the voltage change caused by adjusting one coil turn, denoted as *U_C_*. The equivalent voltage change, ∆u, due to the intermediate plate’s displacement ∆d is expressed as the sum of the quantified equivalent zero-setting voltage, ${nU}_{C}$, and the actual output voltage, $U_{O}$, after zeroing:(3)$$∆u{=nU}_{C}+U_{O}$$

The displacement ∆d is linearly related to ∆u, with *S* representing the sensitivity of the measurement system.(4)∆d=S∆u

### 2.2. Design of the MOSFET Multi-Channel Switch

The core of achieving full-range measurements lies in the system’s ability to automatically select the appropriate transformer tap connection point based on voltage variations in the bridge output, thereby automatically zero-setting the measurement bridge. However, manufacturing a transformer with 10,000 individual turns and controlling 10,000 discrete tap levels is extremely challenging. In practical applications, a multi-stage transformer, cascaded to form a ratio transformer, is commonly used as an equivalent substitute. The structure, illustrated in Figure 5, consists of four stages of 10-turn transformers arranged in cascade. The excitation signal decreases by a factor of 1/10 for each stage. Each stage’s transformer connects to the next through a multi-switch array that selects different taps, ultimately achieving a voltage division accuracy of 0–9999/10,000, equivalent to the voltage division effect of a 10,000-tap transformer. The serial number of the tap connection point *N* corresponds to the number of turns N_2_ described in Section 2.1 and can be expressed as *N = n*_1_
*×* 1000 *+ n*_2_
*×* 100 *+ n*_3_
*×* 10 *+ n*_4_, where *n*_1_ to *n*_4_ represent the selected taps at each stage.

This study designs a bidirectional electronic switch array, controllable via a microprocessor. The switch array consists of 10 electronic switches per group, with four groups installed at the positions indicated in Figure 5, to select the taps of the ratio transformer at various stages. The structure of a single switch is shown in Figure 6, comprising a series connection of NMOS and PMOS transistors configured head-to-tail alongside a driving circuit. The drain of the NMOS transistor connects to the tap of the previous transformer stage, while the source of the PMOS transistor connects to the next transformer stage. This configuration functions equivalently to the controllable switch structure depicted on the right side of Figure 6.

The measurement bridges of strainmeters typically use ±50 V or higher AC signals as the excitation source. Consequently, the source terminals of the MOS transistors must withstand voltage fluctuations of at least ±5 V. To prevent unintentional conduction of the electronic switches, this paper implements a bipolar saturation drive control method. When the microcontroller outputs a low-level signal, the bipolar drive circuit applies −12 V to the gate of the NMOS transistor and +12 V to the gate of the PMOS transistor. Despite source voltage fluctuations, both MOS switches maintain the correct Vgs bias, ensuring they remain in the off state. Under these conditions, the NMOS transistor blocks the positive half-cycle of the AC excitation signal, and the PMOS transistor blocks the negative half-cycle, effectively disconnecting the tap from the subsequent transformer stage. Conversely, when the microcontroller outputs a high-level signal, the drive circuit applies +12 V to the gate of the NMOS transistor and −12 V to the gate of the PMOS transistor, ensuring proper Vgs bias. Both MOS transistors conduct simultaneously, selecting the tap and connecting it to the next transformer stage.

### 2.3. Design of the Automatic Zero-Setting Control System

The automatic zero-setting control system consists of two main components: the upper computer and the lower computer, which communicate via a bus. The primary function of the lower computer is to manage the switch array and select the specified tap connection point of the ratio transformer. The primary function of the upper computer is to calculate the real-time position of the sensor’s plate and the appropriate tap positions, *N*, by monitoring feedback from the sensor’s output voltage, *U_O_*, and maintaining the measurement bridge in a near-balanced state by commanding the lower computer. The workflow is shown in Figure 7.

The lower computer, which is built on the microcontroller, sends the control signals to drive four groups of 10 electronic switch arrays, selecting the tap connection point *N* based on a command from the upper computer. The method use to connect the switch array and the ratio transformer is shown in Figure 5 in Section 2.2. While the measurement bridge maintained a near-balanced state, the voltage signal output from the sensor was processed by the strain host circuit and then collected by the ADC as the signal *U_O_*, which is transmitted to the upper computer.

The upper computer, which is an ARM-based Linux embedded system, calculates the equivalent voltage representing the sensor’s displacement in real-time using the current *N* value and *U_O_*, based on Equation (3) in Section 2.1. If the input imbalance voltage *U_O_* exceeds the preset zero-setting threshold, the automatic control algorithm calculates the target tap serial number $N_{aim}$, as given in Equation (5), and sends commands to the lower computer through a delay control algorithm to adjust the tap of the ratio transformer to the target value, thereby rebalancing the bridge.(5)$$N_{aim}=N_{current}+\frac{U_{O}}{U_{C}}$$

Here,$\;N_{current}$ represents the current tap serial number, *U_O_* is the feedback voltage from the ADC, and *U_C_* is the voltage change caused by a single change in the transformer tap, as described in Equation (3) from Section 2.1. The ratio *U_O_*/*U_C_* determines the adjustment step size. For instance, if *U_C_* = 150 mV and *U_O_* = 1500 mV, adjusting *N* by 10 steps will bring the measurement bridge back to a balanced state. In this system, the maximum ADC output *U_O_* is ±1500 mV; when *U_O_* exceeds the circuit range, the ADC outputs the maximum value, making it impossible for the system to accurately calculate the required adjustment step for the zero-setting. At this time, continuous adjustments are needed until it falls back within the required range. To prevent the adjustments from exceeding the measurement range, the system sets a maximum adjustment step size of approximately 20 coil turns.

Due to the filtering circuits, a stabilization period is required after each adjustment, introducing a delay in bridge control. To ensure rapid and precise zeroing without oscillation when the ADC output exceeds its range, the system incorporates a delay control algorithm that dynamically adjusts the adjustment frequency between 0.5 and 5 s. The flowchart of the delay control algorithm is shown in Figure 8.

When the input signal *U_O_* exceeds the maximum range, the system adjusts the tap connection point with the largest step size and fastest frequency. Once the signal returns within the range, the system reduces the step size and dynamically adapts the stabilization time based on the adjustment’s magnitude. This design enables the system to rapidly zero the bridge, achieving an adjustment speed of up to 20 steps per second. For smaller adjustments, the bridge can reach a balanced state within 1–2 steps.

Step changes with varying magnitudes can be applied to the sensor’s plate to simulate the significant crustal deformation that accumulates over the long-term operation of the measurement system. The zero-setting process and resulting measurement curves are illustrated in Figure 9.

The left graph shows the measurement curve for a single measurement range when the plate undergoes a step change of 0.6 μm at 0 s. Due to the filtering circuits, the system stabilizes after approximately 12 s. In this scenario, the voltage does not exceed the zero-setting threshold and the automatic zero setting is not triggered. The right graph illustrates the measurement curves for plate displacements of 2 μm, 11 μm, and 21 μm. After the ADC output voltage exceeds the zero-setting threshold, the system rapidly zeroes the bridge, quantifying the zero-setting process as the equivalent voltage output and recording the complete measurement curve. The system achieves a dynamic range equivalent to 100 V. At t1 = 28 s, the zero-setting time for a 2 μm step change is recorded. Similarly, t2 = 42 s corresponds to an 11 μm step change, and t3 = 58 s corresponds to a 21 μm step change. These results demonstrate that the system responds quickly and can zero step changes within 20 μm in under 1 min.

## 3. Experiments and Testing

### 3.1. Experimental Platform Introduction

The experimental platform consists of three main components—the sensor, the RZB strainmeter host, and the automatic control and data acquisition system—as shown in Figure 10. The differential capacitive sensor is fixed on a micromotion measurement table, which applies displacements with an accuracy of 0.1 µm. The RZB strainmeter host is connected to the sensor via a cable and includes a ratio transformer, AC amplification, phase-sensitive detection, and low-pass filtering circuits, together with an excitation generation circuit. An electronic switch array is installed within the RZB strainmeter host, replacing the original mechanical dial-switch. The automatic zero-setting control system is integrated with the ADC in the data acquisition system, enabling communication with the host via a digital bus. The entire experimental setup is placed on a seismic isolation test platform.

### 3.2. Quantitative Calibration of the Zero-Setting Experiment

In Equations (3) and (4) of Section 2.1, using *U_C_* as an intermediate variable, the relationship between the tap point *N* of the ratio transformer, the output voltage *U_O_*, and the measured displacement is established. To verify the linear quantification capability of the automatic zero-setting process based on electronic switches and to calibrate the voltage change, *U_C_*, associated with a single-turn adjustment of the transformer coil, standard test components were integrated into the measurement system under controlled laboratory conditions. Once the system output stabilized, an electronic switch was used to adjust one turn of the transformer coil, and the corresponding ADC output voltage values were recorded until the signal approached the full-scale range of the ADC. The experimental data are shown in Table 1, and the fitting curve is shown in Figure 11. The voltage change, *U_C_*, induced by one coil turn adjustment was determined to be 145.16 mV, with a nonlinearity error of 0.912%.

### 3.3. Linearity Calibration Experiment

To assess the linearity of the system over its full measurement range, the RZB strainmeter was used as the experimental platform. A differential capacitive sensor was fixed on a micro-motion measurement stage, adjusted to within the effective linear range, and the measurement bridge was balanced. Displacements of up to 86 μm were applied cumulatively to the sensor by turning the micro-motion stage knob. The system tracked these displacements, automatically zero-setting and outputting equivalent voltage values. The experimental data are shown in Table 2, and the calibration curve, fitting curve, and residuals are shown in Figure 12.

The experimental results demonstrate that the displacement step applied by the micro-displacement stage is approximately 1 µm, and the equivalent voltage change output of the measurement system is about 2862 mV, which corresponds to the sensitivity S in Equation (4). The linearity error was calculated using Equation (6), and according to the Industry Standards(DB/T 31.2-2008) [20]; this value should be less than 1%.(6)$$Linearity\;error=\frac{\left|{∆y}_{max}\right|}{y_{Full\;Scale}}\times 100\%$$

In this Equation, $\left|{∆y}_{max}\right|\;$= 676.9 mV is the maximum absolute error and $y_{Full\;Scale}\;$= 246,242.868 mV is the system output range. The linearity error of the measurement system is 0.27%, which already meets the industry standard requirement of less than 1%. Analyzing the distribution of residuals reveals a clear regularity; it was confirmed that this regularity is due to systematic errors rather than being introduced by the electronic switch array through extensive repeated experiments and comparative experiments. Theoretically, these errors could be predicted and corrected using mathematical methods. The experimental results demonstrate excellent linearity over the 86 µm range, with a nonlinearity error of 0.27%, fully utilizing the sensor’s effective linear range [19].

### 3.4. Noise Testing and Resolution Testing

To evaluate the noise level of the electronic system and its theoretical resolution, standard test components were connected to the measurement system under laboratory conditions. Once the system output was stabilized, an output signal curve was recorded, as shown in Figure 13, with a sampling period of 0.5 s and 2740 sampling data points.

The theoretical minimum resolution was calculated using the ratio of the effective value of the measured output noise $y_{n}$ to system *sensitivity S*:(7)$$y_{n}=\sqrt{\frac{1}{N}\sum_{i=1}^{N}(y_{i}-\bar{y}{)}^{2}}$$(8)$$R\left(Resolution\right)=\frac{y_{n}\left(Effective\;value\;of\;noise\right)}{S\left(Sensitivity\right)}$$

Given $y_{n}\;$= 0.00391 mV, a peak-to-peak noise of 0.024 mV, and a system *sensitivity S* of 2862.5755 mV/μm (Figure 12), the theoretical minimum *resolution R* was calculated as 0.00136 nm.

## 4. Conclusions

This study designed a range-extension measurement system utilizing MOSFET-based electronic multi-switches and an automatic zero-setting control approach. The system integrates a Linux-based embedded upper computer with a microcontroller to control MOS bidirectional switches for transformer tap selection. It achieves automatic zero-setting for a differential capacitive sensor measurement bridge and enables full-range measurements by quantifying zero-setting actions into equivalent voltage values.

(1)The system successfully implements automatic zero-setting and full-range measurement, reducing data loss risks associated with traditional manual zero-setting methods. Its fully electronic design enhances its integration and stability.(2)Laboratory testing showed an electronic noise level below 0.024 mV and a theoretical minimum resolution of 0.00136 nm. Zero-setting quantification calibration achieved a linearity error of 0.91% and large-range linearity calibration with a differential capacitive sensor yielded a nonlinearity error of 0.27%. The system can zero large step changes of up to 20 μm within 1 min and record the full measurement curve.(3)The system design incorporates modularity and compatibility, facilitating ease of human–machine interaction and ensuring compatibility with legacy equipment. It can serve as a solution for automated full-range measurements or as an upgrade to traditional manual dial systems, particularly in applications involving crustal stress and strain measurements.

## Figures and Tables

**Figure 1 sensors-25-00476-f001:**
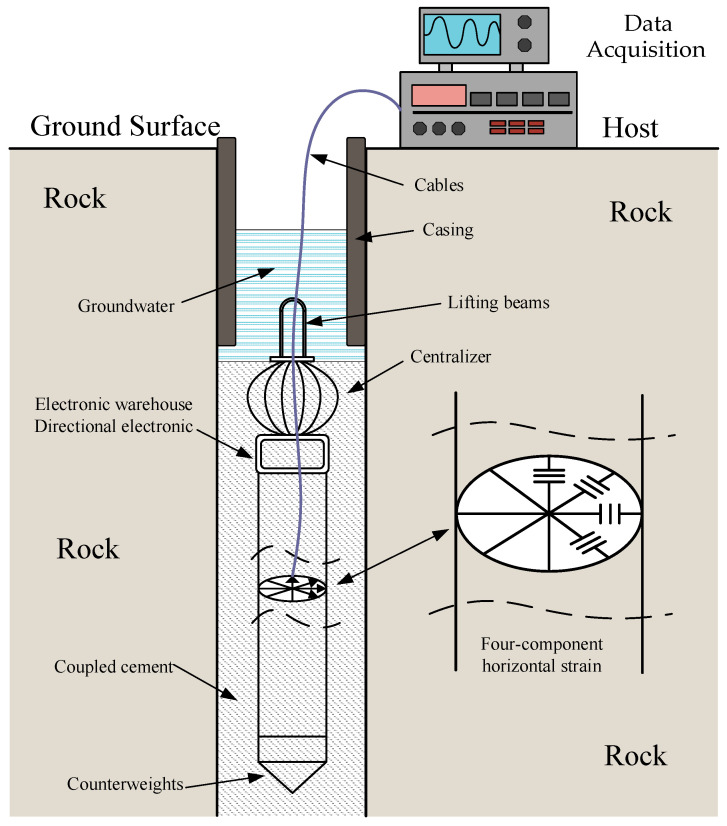
Schematic diagram of borehole strain observation system.

**Figure 2 sensors-25-00476-f002:**
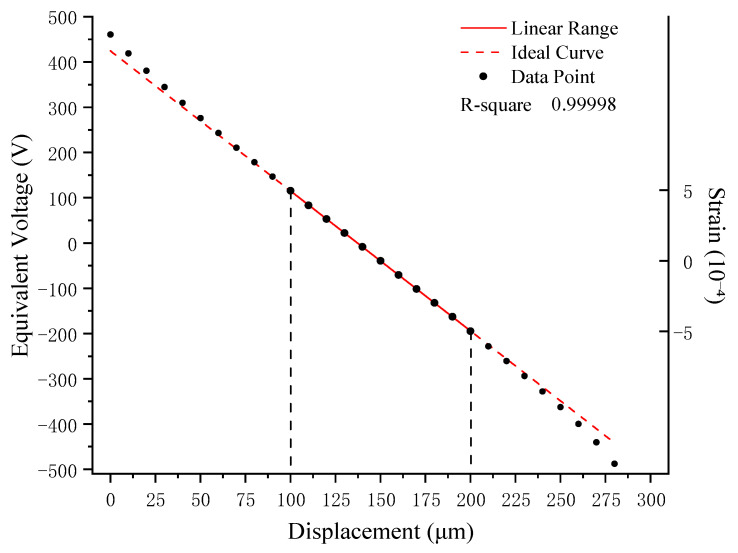
Calibration curve for the full range of the sensor.

**Figure 3 sensors-25-00476-f003:**
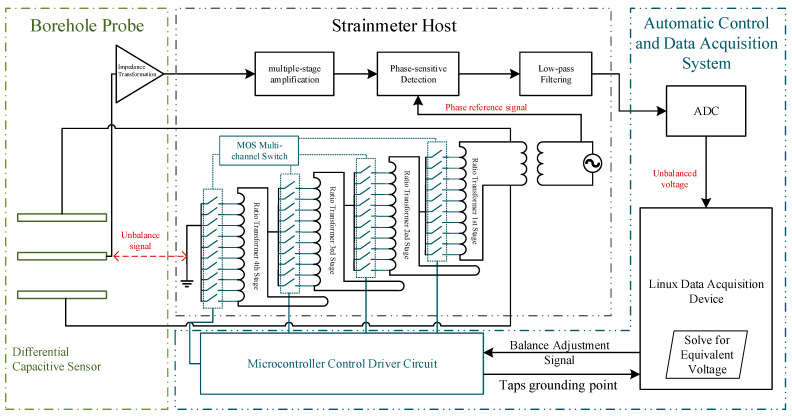
Design of a full-range measurement system based on the automatic zero setting.

**Figure 4 sensors-25-00476-f004:**
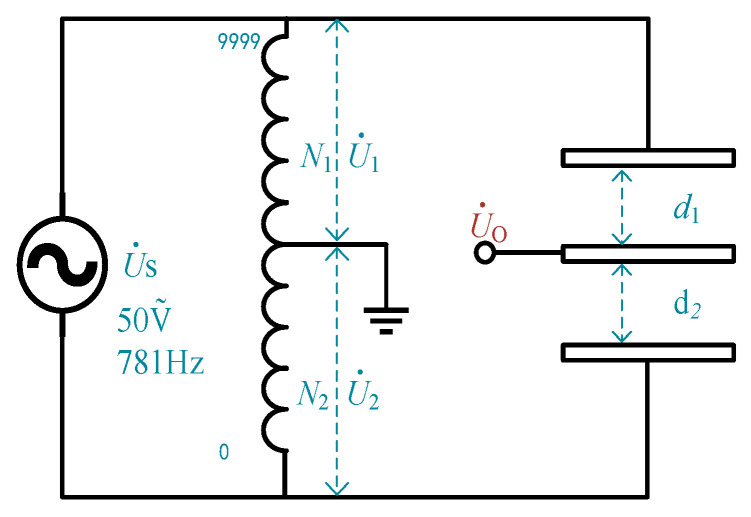
Measurement bridge equivalent circuit diagram.

**Figure 5 sensors-25-00476-f005:**
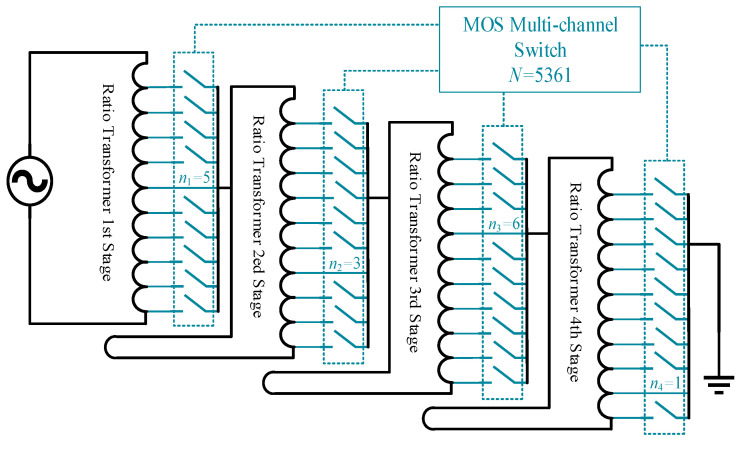
Ratio transformer and multi-switch array structure diagram.

**Figure 6 sensors-25-00476-f006:**
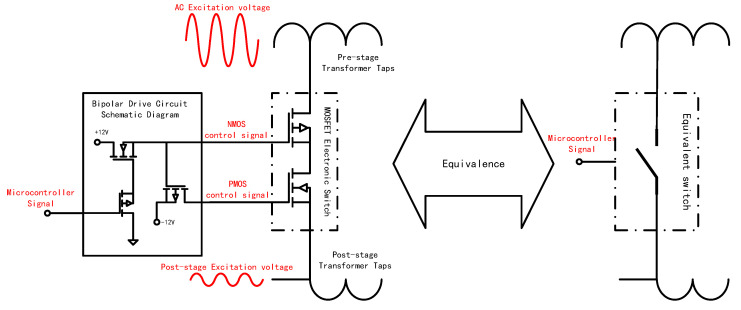
MOSFET bidirectional switch schematic.

**Figure 7 sensors-25-00476-f007:**
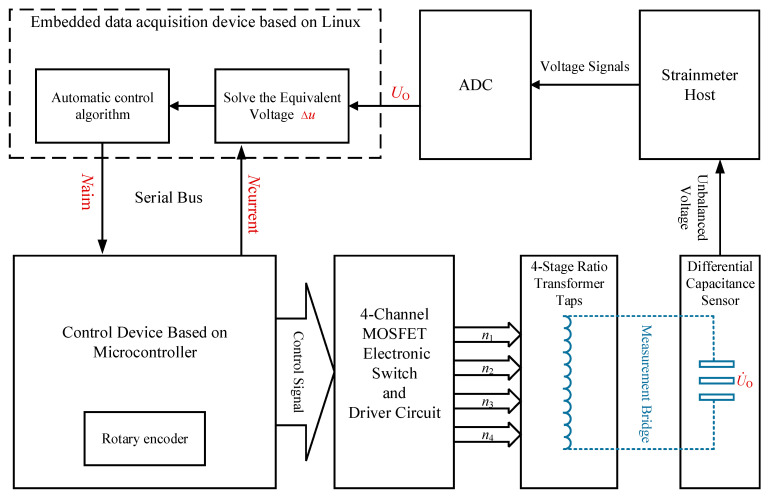
Automatic zero-setting control system block diagram.

**Figure 8 sensors-25-00476-f008:**
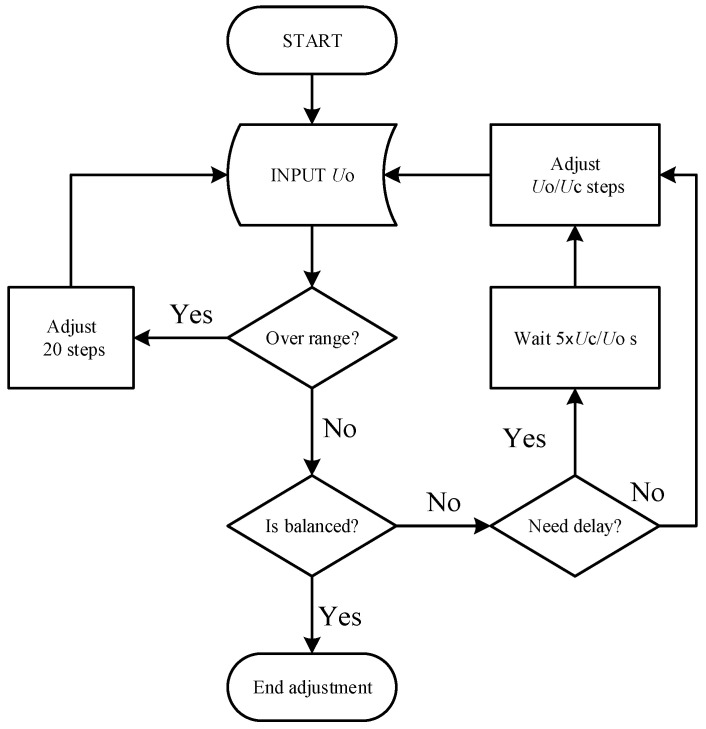
Delay control algorithm flowchart.

**Figure 9 sensors-25-00476-f009:**
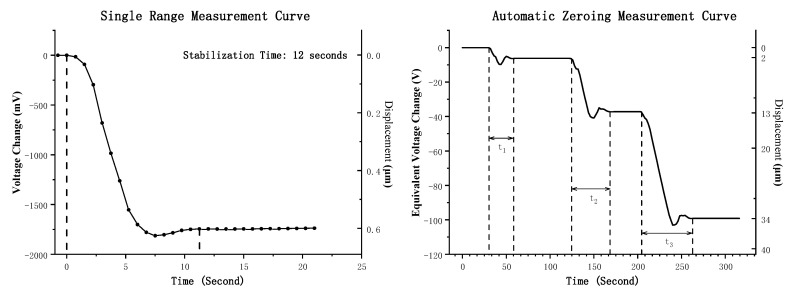
Automatic Zero Setting Measurement Curve.

**Figure 10 sensors-25-00476-f010:**
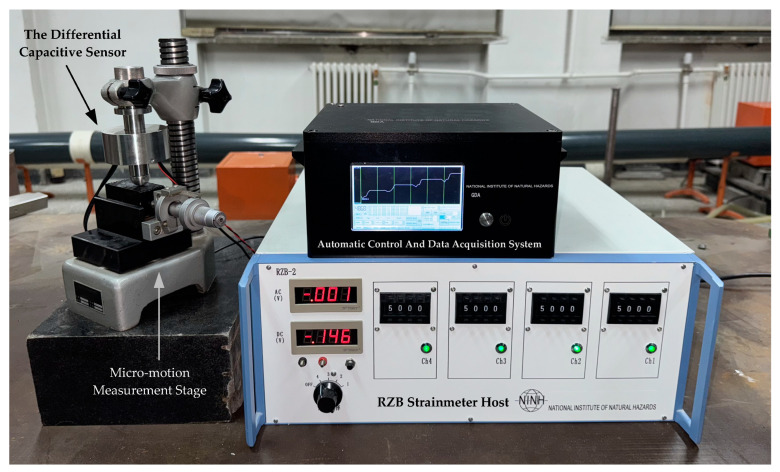
Full-range measurement system based on the automatic zero setting.

**Figure 11 sensors-25-00476-f011:**
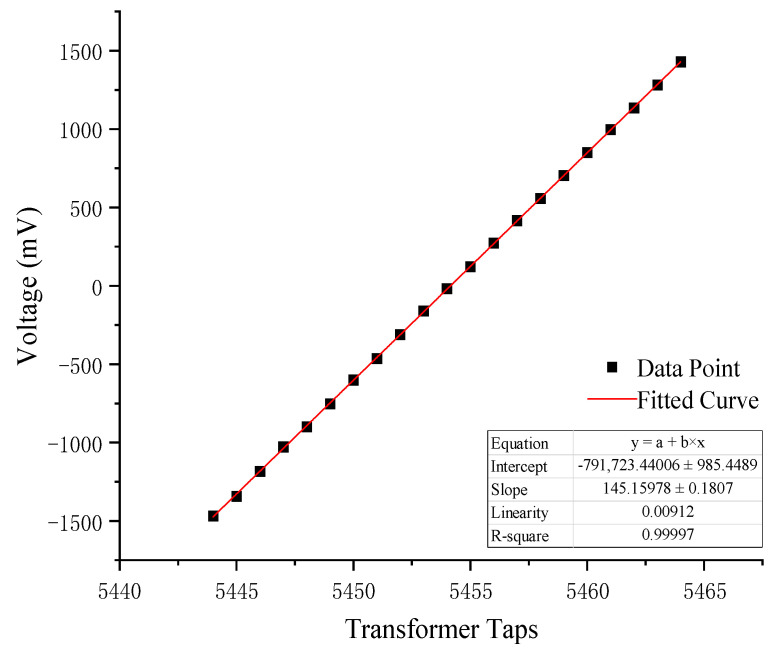
Quantitative calibration curve.

**Figure 12 sensors-25-00476-f012:**
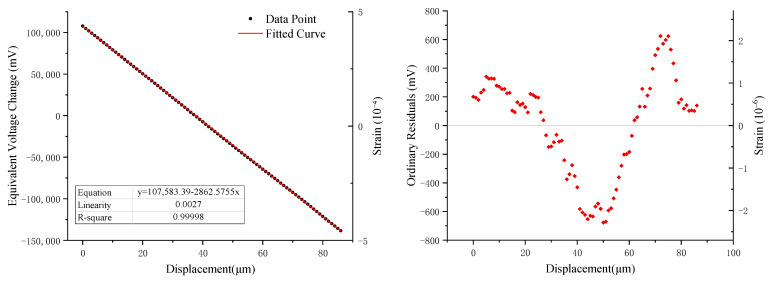
Linearity calibration curve.

**Figure 13 sensors-25-00476-f013:**
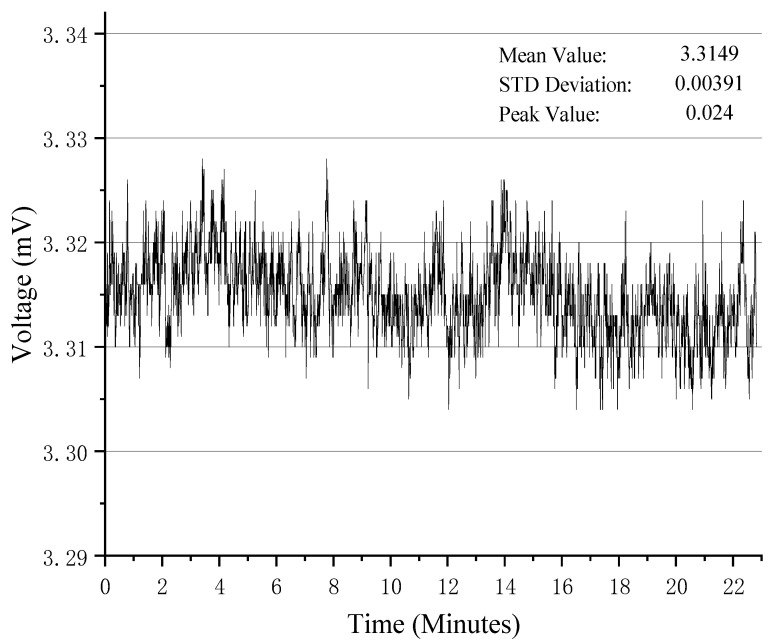
System output noise curve.

**Table 1 sensors-25-00476-t001:** Quantitative calibration test data.

*N*	Voltage (mv)	*N*	Voltage (mv)
5444	−1469	5455	122
5445	−1343	5456	273
5446	−1183	5457	416
5447	−1028	5458	556
5448	−900	5459	704
5449	−752	5460	850
5450	−601	5461	997
5451	−463	5462	1136
5452	−312	5463	1281
5453	−160	5464	1429
5454	−18	5455	122
5444	−1469		

**Table 2 sensors-25-00476-t002:** Linearity calibration experiment data.

Δd (μm)	ΔV (mv)	Err (mv)	Δd (μm)	ΔV (mv)	Err (mv)	Δd (μm)	ΔV (mv)	Err (mv)	Δd (μm)	ΔV (mv)	Err (mv)
0	107,784.111	200.724	22	44,827.505	220.779	44	−19,023.257	−653.322	66	−81,215.114	131.482
1	104,915.231	194.419	23	41,957.199	213.049	45	−21,862.209	−629.698	67	−83,999.909	209.263
2	102,037.514	179.278	24	39,081.193	199.618	46	−24,729.664	−634.578	68	−86,813.207	258.540
3	99,225.926	230.265	25	36,213.737	194.738	47	−27,523.011	−565.349	69	−89,538.145	396.178
4	96,381.843	248.758	26	33,249.370	92.946	48	−30,364.813	−544.575	70	−92,305.838	491.060
5	93,611.300	340.790	27	30,330.608	36.760	49	−33,263.622	−580.809	71	−95,124.837	534.636
6	90,735.293	327.360	28	27,363.391	−67.882	50	−36,222.288	−676.900	72	−97,898.231	623.818
7	87,873.539	328.181	29	24,418.976	−149.721	51	−39,078.342	−670.379	73	−100,814.142	570.482
8	85,008.934	326.151	30	21,560.072	−146.050	52	−41,863.137	−592.598	74	−103,650.244	596.956
9	82,098.723	278.516	31	18,726.821	−116.726	53	−44,710.640	−577.526	75	−106,486.345	623.430
10	79,228.418	270.786	32	15,916.372	−64.599	54	−47,503.986	−508.296	76	−109,442.161	530.190
11	76,349.561	254.505	33	13,006.162	−112.234	55	−50,305.884	−447.618	77	−112,400.828	434.099
12	73,487.806	255.325	34	10,150.108	−105.712	56	−53,082.128	−361.286	78	−115,382.298	315.205
13	70,594.698	224.792	35	7151.536	−241.708	57	−55,864.072	−280.656	79	−118,400.822	159.256
14	67,735.793	228.464	36	4155.815	−374.854	58	−58,648.867	−202.875	80	−121,239.773	182.880
15	64,748.623	103.869	37	1328.265	−339.829	59	−61,507.772	−199.204	81	−124,167.086	118.143
16	61,875.467	93.288	38	−1470.782	−276.300	60	−64,355.275	−184.131	82	−127,006.038	141.766
17	59,082.121	162.518	39	−4409.496	−352.439	61	−67,105.865	−72.147	83	−129,907.697	102.682
18	56,200.414	143.386	40	−7351.061	−431.428	62	−69,859.307	36.988	84	−132,766.602	106.353
19	53,347.210	152.758	41	−10,363.884	−581.676	63	−72,701.109	57.761	85	−135,634.057	101.474
20	50,459.803	127.926	42	−13,251.292	−606.508	64	−75,488.754	132.691	86	−138,458.757	139.349
21	47,560.994	91.692	43	−16,130.149	−622.789	65	−78,227.944	256.077			

## Data Availability

Data are contained within the article.

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
