# Peer review of "Range Extension of Borehole Strainmeters Using MOSFET-Based Multi-Switch Automatic Zero Setting"

_sensors, 2025, doi:10.3390/s25020476_

Round 1

Reviewer 1 Report

Comments and Suggestions for Authors

This paper presents a full-range measurement system was developed using the a bidirectional analog multi-switch based on MOS transistors and automatic feedback control, it provides a new method for borehole in-situ stress testing. However, some comments have been provided and the authors should consider them to improve the manuscript. I recommended accepting the manuscript with minor revisions.

1.     The manuscript should show photos of the full-range measurement system, as well as photos of the test system;

2.     In order to facilitate the understanding of scientific research and engineering personnel, it is necessary to explain how the full-range measurement system implements borehole in-situ stress testing;

3.     References should be ordered in the order in which they appear, such as the reference number for the “Borehole strainmeters play a critical role in crustal deformation monitoring ”should be [2,3], not [2,17];

4.     The format of the references needs to be unified.

Author Response

Dear Reviewer and Editor:

I hope this message finds you well. I would like to express my sincere gratitude for taking the time to review my manuscript <sensors-3399178>. Your valuable insights have significantly contributed to enhancing the quality of our research and have provided important guidance for revising and improving our paper. We have carefully considered your comments and made the necessary revisions, and we hope these changes will meet your approval.

Below are my responses to the specific issues you raised:

Q1. The manuscript should show photos of the full-range measurement system, as well as photos of the test system;

Thank you for your suggestions. Adding photos of the full-range measurement system will help readers understand the experimental process. We have included photos and descriptions of the experimental platform in the text.

As shown in the figure, the measurement system mainly consists of the strain gauge main unit and the data acquisition control system. The differential capacitive sensor in the strain gauge probe is fixed on a micromovement measurement platform, allowing specific displacement amounts to be applied.

The figure also depicts the calibration experiment mentioned in the text, where the curve displayed on the screen of the data acquisition control system represents the automatic zeroing measurement curve when a 1µm step change is applied to the sensor.

Q2. In order to facilitate the understanding of scientific research and engineering personnel, it is necessary to explain how the full-range measurement system implements borehole in-situ stress testing;

We have added a schematic diagram of the RZB strain gauge used in field measurements in the first chapter of the text. During field measurements, the strain probe needs to be installed in a borehole well that is several hundred to one thousand meters deep. The probe's steel cylinder casing is coupled to the bedrock of the borehole using expanding cement. Inside the probe, four differential displacement sensors are installed to measure the relative deformation in four directions of the borehole. These sensors are connected via cables to the main unit and data acquisition system on the surface. The surface equipment simultaneously resolves the data from the four sensors to calculate the relative principal strains and angles in the borehole.

It should be noted that, the RZB type of component strain measurement device can only measure stress changes after the equipment is installed; it does not measure the in-situ absolute stress of the borehole. Therefore, in our routine data processing, we use a specific observation value as a reference to calculate the relative differences of other observation values

The specific measurement principle is as follows: for two-dimensional stress and strain changes, the circular steel casing of the probe is abstracted as a circle. When subjected to forces from different directions, the circle undergoes deformation and transforms into an ellipse. This change can be described using three parameters: the lengths of the major and minor axes of the ellipse, as well as the orientation of the major axis. To quantify the degree of deformation, we establish the following definitions:

ε1 = ?−?/ ?, ε2 = ?−?/r

Sθ=A(ε1+ε2 )+B(ε1-ε2 ) cos(θ-φ)

Here,  is the angle of the sensor arrangement,  is the angle of the maximum principal strain, A and B are related to the effective Young's modulus and Poisson's ratio of the cylinder and surrounding rock, and the relative deformation  is measured by the sensor. This equation has three unknowns, ε1, ε2, and φ. By solving the simultaneous equations from four sensors, the relative deformations in the uniform horizontal principal strains ε1 and ε2, as well as the direction φ, can be determined.

Q3. References should be ordered in the order in which they appear, such as the reference number for the “Borehole strainmeters play a critical role in crustal deformation monitoring” should be [2,3], not [2,17];

Q4.  The format of the references needs to be unified.

Thank you very much for carefully pointing out the issues. The reference format and citation order have been revised in the text. We have checked the format of the references, authors' names, and related information against Google Scholar and have made the necessary corrections in the text. The revised results are as follows:

We hope you can recognize our conclusions. If you have any further questions, please feel free to contact us through the editorial office at any time!

Reviewer 2 Report

Comments and Suggestions for Authors

The paper "Range Extension of Borehole Strainmeters Using MOSFET-Based Multi-Switch Automatic Zero setting" is written fine, however, there are some comments:

1. Explain why inductances are used instead for example the capacitances.

2. Referring to Figure 3 and the equations that describe the operation of the bridge, the frequency and amplitude of signal Us should be clearly specified together with the information about that selection.

3. The capacitors have some associated parasitic capacitances that are not considered.

4. What is the accuracy of transformer ratio

5. The switch resistances may be important in noise and transients

6. Explain which control algorithm is used (PID with parameters...)

7. The explanation of Figure 6 block diagram operation is not sufficiently detailed.

8. The calibration curve in Figure 9 shows linear results; in reality, also the displacement from the linear curve would be necessary to show on the same picture

9. Table 2 shows linearity calibration experimental data. The deviation from the ideal curve should be presented as well

10. Figure 10 should include deviation from the ideal calibration curve

11. Compare the results of that investigation to the results from the references.

Author Response

Dear Reviewer and Editor:

I hope this message finds you well. I would like to express my sincere gratitude for taking the time to review my manuscript <sensors-3399178>. Your valuable insights have significantly contributed to enhancing the quality of our research and have provided important guidance for revising and improving our paper. We have carefully considered your comments and made the necessary revisions, and we hope these changes will meet your approval.

Below are my responses to the specific issues you raised:

Q1. Explain why inductances are used instead for example the capacitances.

The differential capacitive micro-displacement sensor is the core sensor of the strain measurement system. The main role of the inductance in the measurement system is to form a ratio arm with the differential capacitive displacement sensor, thereby creating a Wheatstone bridge. The capacitor is directly coupled with the measured object, while the inductance (ratio-transformer) does not participate directly in the measurement. Its primary function is to adjust the turns ratio, thereby modifying the bridge's voltage divider ratio to balance the bridge circuit. The reasons for selecting an inductance to form a ratio bridge with the differential capacitor are as follows:

  1. High controllability and ease of fabrication: The reactance of the inductor is mainly related to the number of coil turns in the transformer, allowing for the production of a transformer with a stable voltage divider ratio by winding the appropriate number of turns. However, it is very challenging to produce resistors or capacitors with highly consistent reactance.
  2. Good linearity: The turns ratio of the transformer and the voltage divider ratio exhibit high linearity, with minimal marginal effects.
  3. Low internal resistance for low-frequency signals: The low internal resistance of the inductor is beneficial for high-precision measurements.

For specific principles of high-precision ratio arm measurement, please refer to the paper:

< Ouyang Zuxi, Zhang Zongrun, He Chengping. A High-Precision Displacement Measurement System Based on Inductive Coupling Ratio Arm [J]. Application of Electronic Technology, 2004, 30(010):41-43. DOI: 10.3969/j.issn.0258-7998.2004.10.015.>.

Due to the extremely small short-term variations in ground strain, which are nearly quasi-static measurements, the output signal from the capacitive sensor is very weak, typically requiring amplification by about 100 times to meet the required measurement precision. However, the long-term cumulative changes in ground strain are significantly larger compared to daily variations. After prolonged operation of the measurement system, excessive displacement of the capacitive sensor's plates can lead to issues such as signal limitation in subsequent processing circuits and exceeding the range of the ADC acquisition circuit, resulting in data loss. Therefore, it is necessary to adjust the tap grounding point of the inductance to modify the voltage divider ratio of the Wheatstone bridge, bringing the output voltage of the plates back to a near-balanced state.

In practical measurements, the measurement bridge usually requires rebalancing every 1-2 months, and in areas with significant geological changes, the adjustment intervals are even shorter. The traditional adjustment method involves grounding the transformer taps via mechanical switch rotary dials, where operators manually change the grounding sequence on the dial to adjust the turns. However, many monitoring points are installed in outdoor environments, and failure to adjust in a timely manner can lead to data accuracy issues. This highlights the significance of this research: by designing a controllable electronic switch to replace the original mechanical switch, combined with an automatic control system, we aim to achieve automatic zero-setting functionality, thus enabling full-range measurements and reducing the risk of data loss.

Q2. Referring to Figure 3 and the equations that describe the operation of the bridge, the frequency and amplitude of signal Us should be clearly specified together with the information about that selection.

The currently used RZB strainmeter employs an excitation signal Us of 50V at a frequency of 781Hz. We have noted the parameters of the excitation signal in the text and provided a brief explanation according to your suggestions.

The reason for selecting 50V is that the gap between the plates of the capacitive sensor is approximately 5×10-4/m, and the industry standard:

< DB/T 31.2—2008; Technical requirements of instruments in network for earthquake monitoring.The instrument for crustal deformation observation. Part 2: Strainmeter (in Chinese)> , requires a minimum measurement resolution of 5×10-11/m, which is one ten-millionth of the total range. If the excitation voltage is too low, it will result that the signal-to-noise ratio is insufficient for accurate signal discrimination during precise measurements. In theory, a higher excitation voltage is better; however, in practical measurements, increasing the voltage can highlight marginal effects, and excessively high voltages can also affect system stability. Therefore, 50V has been established as an empirical value based on long-term operation. The choice of 781Hz is based on the following reasons:

  1. Slow Short-Term Changes in Ground Strain: For important daily monitoring targets, the period of solid tide changes is 24 hours, while the sampling frequency of the measurement system is typically set to one minute. In order to improve measurement accuracy, the low-pass filtering circuit in the strainmeter host is usually configured with a filter time of over 60 seconds, which can be considered as quasi-static measurement. Therefore, a high excitation frequency is unnecessary, and the excitation frequency should be minimized within the range that ensure the linearity of the sensor's capacitive characteristics.
  2. Installation Depth of Borehole Strainmeters: The borehole strainmeter needs to be installed in deep wells ranging from several hundred to 1000 meters underground, while the strainmeter host, including the ratio transformer, excitation circuit, and processing circuit, is located on the surface. The excitation signal of the ratio bridge and the output signal from the sensor must be transmitted over long distances via cables. Using a lower frequency excitation signal can effectively reduce interference, minimize the impact of parasitic capacitance, and enhance transmission distance. Adopted this layout because the sensors are embedded in coupling cement underground, making maintenance impossible. Hence, the underground structure should be as simple and reliable as possible, while the maintenance-required components are placed on the surface. This also highlights the advantages of using an electrical zero-setting scheme with the ratio transformer compared to integrating a mechanical zero-setting structure in the underground sensor.
  3. Measurement of Sensor Signals: The measurement of the sensor signals is primarily achieved through amplification, phase-sensitive detection, and low-pass filtering to obtain the effective DC component. Therefore, the frequency of the excitation signal does not significantly affect measurement accuracy; in fact, excessively high frequencies can increase errors in phase-sensitive detection. Compared to frequency, we are more concerned with the stability of the excitation signal's amplitude, as it is directly related to the effective DC component of the signal.

In summary, based on long-term operational experience, we have selected a working frequency of 781Hz. This frequency is an integer division of the crystal oscillator commonly used in microcontrollers, facilitating device development and maintenance.

Q3. The capacitors have some associated parasitic capacitances that are not considered

We sincerely apologize for not considering the impact of parasitic capacitance on the system in this study. We would like to explain this issue from the following two aspects:

  1. The RZB type of component strain measurement device can only measure strain changes relatively; it does not measure the in-situ absolute strain of the borehole. Therefore, in our routine data processing, we use a specific observation value as a reference to calculate the relative differences of other observation values. Since the measurement process is nearly quasi-static, we believe that the impact of parasitic capacitance on the system is also stable, thus theoretically this stable deviation has been eliminated during the calculation of relative differences.
  2. According to the response to Q2, the excitation signal and operating frequency of the system are very low; therefore, we believe that the parasitic capacitance effects at this frequency are negligible.

In summary, we rarely consider the influence of parasitic capacitance in practical work. However, we greatly appreciate your valuable suggestions, as they provide important insights for our further research into the equipment, expanding its functionality, and improving its performance.

Q4. What is the accuracy of transformer ratio

According to the response to the Q1, the ratio transformer is an important component of the measurement bridge, forming a high-precision ratio measurement arm together with the differential capacitor. The transformer also has advantages such as high linearity and stable voltage divider ratios. In this study, the RZB transformer uses an equivalent 10,000-turn tap transformer, which can achieve a voltage division accuracy of 1/10,000 for the excitation voltage Us​. Based on the principle of relative measurement in a ratio bridge, the adjustment of one turn in the transformer causes an unbalanced output voltage Uc in the bridge, which is equivalent to a change of approximately 50nm for a relative displacement of the capacitor sensor plates over a total distance (usually 0.5mm) of 1/10,000. This voltage is amplified, detected, and filtered by the host circuit, resulting in an effective output voltage adjustment of about 145mV per turn.

In summary, the adjustment precision of the ratio transformer in the experimental equipment platform is 145mV/50nm. Since this is a relative measurement, the value may vary slightly due to differences in excitation voltage and the gap between sensor plates in different devices, necessitating precise calibration for each set of equipment. It is important to note that this precision pertains to the adjustment of the ratio transformer in the balancing bridge, rather than the overall measurement precision of the system, which depends on the accuracy of the capacitive sensor and the measurement circuit in the RZB host. We focus more on the stability and linearity of the transformer’s voltage division, as well as the overall measurement resolution of the equipment. The widely used RZB type strainmeter has achieved a measurement resolution of 5×10−11/m in practical applications, with linearity better than 1%.

The ratio transformer, due to its stable voltage divider characteristics and linearity, is central to measurements and calibrations in the ratio measurement system. For specific principles of measurement and calibration, please refer to the paper:

 <Qiu Z H, Shi Y L, Ouyang Z X. Relative in-situ calibration of 4-component borehole strain observation[J]. Journal of Geodesy and Geodynamics, 2005, 25(1): 118-122.>.

Furthermore, after introducing new electronic switching devices into the existing RZB framework in this study, we aims to maintain the original performance of the equipment, as verified in the experimental section of Chapter 3.

Q5. The switch resistances may be important in noise and transients

The switch resistance and noise are important metrics in the design of electronic switches. We apologize for the limited conditions in our laboratory, which prevent us from conducting a quantitative analysis of the electronic switches. However, based on the following two reasons, I believe this aspect of the analysis does not significantly affect the experimental conclusions:

  1. As stated in the responses to Q2 and Q3, our measurement of ground strain deformation is a slowly varying quasi-static process, with a measurement frequency of one minute and a filtering time of approximately 60 seconds. During the measurement, the primary role of the electronic switch is to continuously select taps to provide an appropriate and stable voltage division ratio. This does not require frequent state changes like modulation, and due to the low operating voltage frequency, adjustments are typically needed only about once a month. Data collected before the circuit stabilizes during adjustments are not usable. Therefore, we believe that the transient response and noise of the switch have virtually negligible impact on the measurement results.
  2. Due to our equipment limitations, we are unable to precisely analyze the impedance and noise of individual switches. However, as mentioned in the response to Q4, we are more concerned with whether replacing mechanical switches with electronic ones will adversely affect the voltage division stability of the ratio transformer, the system's measurement performance, and overall noise levels. The experiments in Section 3.2 demonstrate that the voltage division accuracy remains stable after introducing the electronic switch array. Experiment 3.3 shows that the measurement system with electronic switches meets the linearity error requirements set by the industry standards. Experiment 3.4 confirms that after introducing electronic switches, the overall noise level and resolution of the system also meet the industry standards.

To better address your concerns, we conducted supplementary experiments: we measured the noise of the standard measurement device connected to the RZB type measurement system using conventional mechanical switches and compared it to the system noise of the electronic switch setup described in the text. The experimental results are as follows: the left side shows the system noise measured with the electronic switch, while the right side shows the system noise measured with the mechanical switch.

Due to the different power supplies used in the second test and the higher ambient temperature during testing, the mean and average noise values differ. For both measurements, we applied the resolution estimation formula:

R(Resolution)= yn(Effective value of noise)/S(Sensitivity)

The theoretical resolution of the electronic switch system was measured at 0.00136nm, while the theoretical resolution of the mechanical switch system was 0.00482nm. It is evident that the measurement resolution of the electronic switch array and the mechanical switch connected to the RZB host are roughly in the same order of magnitude, and both theoretical resolutions are better than the industry standard. As for the overall resolution of the system, it is primarily related to the mechanical structure of the sensor and the host circuit. This validates that the introduction of the electronic switch system did not introduce new noise, and the overall noise level is consistent with that of the traditional mechanical switch version.

Q6. Explain which control algorithm is used (PlD with parameters...

I apologize for the confusion caused by my previous wording. In fact, we only referenced the principles of PID control but did not implement a true PID control algorithm. The specific reasons are as follows:

According to the response to Q4, the minimum control precision for rebalancing the bridge by adjusting the transformer taps is approximately 145mV/50nm, while the required measurement precision must reach 0.15mV/0.05nm. The control precision is significantly lower than the measurement resolution, making it impossible to use traditional PID control methods for precise bridge balancing. In practical measurements, we only need to adjust the switch to ensure that the sensor output voltage remains within the ADC range. The ADC range is typically ±1.5V, and we consider an appropriate measurement interval to be around ±1V, which usually requires zero-setting only once a month.

Due to the presence of a low-pass filter circuit, any changes in the state of the measurement bridge require at least 6 seconds to stabilize, including both the zero-setting and measurement processes. To prevent circuit oscillation, we set a delay control factor in the design of the automatic zero-setting algorithm to ensure that the bridge has sufficient stabilization time. Additionally, to quickly zero the bridge when the sensor offset is large and to minimize data loss during the adjustment process, we referenced the idea of differential control to dynamically adjust the delay parameters. The general idea is that when the output voltage exceeds the ADC range, it indicates a significant bridge offset, prompting a reduction in delay time to increase the adjustment frequency and quickly return the bridge to a near-balanced state. Once the output voltage returns to the range, we increase the delay time to allow the bridge to stabilize adequately.

Based on your suggestion, we have provided a more detailed introduction to the algorithm flow in the text and removed any references to PID that could lead to misunderstandings.

Q7. The explanation of Figure 6 block diagram operation is not sufficiently detailed.

Based on your suggestions, we have optimized the flowchart to make it more intuitive and readable. We have also reorganized the language in the text to provide a more detailed introduction to the system's operation process.

The general idea is that the lower computer controls the switch array to control the transformer for stable voltage division, while the host computer and ADC collect the bridge output voltage and transmit it to the upper computer. The upper computer monitors the voltage status, calculates the appropriate switch connection sequence number, and commands the lower computer to maintain or change the switch status.

Q8. The calibration curve in Figure 9 shows linear results; in reality, also the displacement from the linear curve would be necessary to show on the same picture

We sincerely apologize for the oversight in our formatting, which caused some errors in the relationships between the figures and tables. The text has been revised, and the current version correctly reflects that Figure 11 (previously Figure 9) corresponds to Table 1. Additionally, we would like to address why displacement fitting data does not appear in the figures:

  1. The main purpose of Experiment 3.2 (Figure 11 and Table 1) is to calibrate the change in output voltage Uc​ caused by adjusting one turn of the transformer, while also verifying the stability of the voltage division. As explained in the response to Q4, the precision and stability of the ratio transformer’s voltage division are key to ensuring stable measurements in the system. Since each measurement system has minor differences, it is necessary to specifically calibrate the voltage change Uc​ caused by adjusting one turn of the transformer. The experimental results indicate that the system's output voltage for one turn of the coil varies stably around 145mV and exhibits good linearity. Since the stability and linearity of the ratio transformer have been validated in previous work, this experimental result confirms that the introduction of the electronic switch array did not introduce additional errors, and the ratio transformer can still ensure stable voltage division performance.
  2. As stated in the response to Q1, the ratio transformer does not directly participate in deformation measurements. In traditional measurement methods, it is necessary to calibrate the relationship between the number of turns N and the displacement through the leveling of the bridge, and then establish the relationship between the number of turns and the voltage change. However, the full-range measurement method proposed in this paper uses the voltage Uc​ from one turn of the coil as an intermediate variable, establishing a direct relationship between the output equivalent voltage and the displacement. Therefore, Experiment 3.3 allows for direct voltage sensitivity calibration of the displacement without needing to calibrate the displacement in relation to the intermediate variable Uc​.

Q9. Table 2 shows linearity calibration experimental data. The deviation from the ideal curve should be presented as well

Q10.Fiqure 10 should include deviation from the ideal calibration curve

Thank you for your valuable suggestions. We have added error terms to the table and included residual plots along with an error analysis. Since Table 2 and Figure 12 (previously Figure 10) pertain to the same experiment, we will provide a unified response to your Q9 and Q10.

Based on the experimental data, the measurement system demonstrated good linearity during the full-range test of 86µm, with a maximum error of approximately 670mV. According to the calculation method outlined in the China Earthquake Industry Standards (DB/T31.2-2008), the linearity of this measurement system is 0.27%, which is better than the industry standard requirement of 1%, thus meeting the measurement standards. However, the error distribution shows notable patterns. Although it falls within the required range, to better address your questions, we conducted two additional sets of control experiments. These experiments used both electronic switches and traditional mechanical switches to recalibrate the sensor over a larger range of 110µm. The experimental results are as follows:

Due to differing environmental conditions and temperatures, as well as variations in the initial positions of the sensors, the results of these two experiments differ slightly from previous findings. Both experiments exhibited linearity better than 0.3%, which complies with industry standards. Comparing the results of the three experiments, it is evident that the system's measurement performance remains high with both mechanical and electronic switches. Additionally, the residual distribution plots for all three sets show similar characteristics. It is particularly worth noting that the experimental platform used for the mechanical switch control group is the currently operational RZB type strain measurement system. Therefore, we can conclude that this error distribution arises from system errors, rather than being introduced by the electronic switches, and can theoretically be corrected mathematically.

In summary, the electronic switch designed in this paper has maintained the performance of the RZB strainmeter while achieving an expansion of the measurement range.

Q11.Compare the results of that investigation to the results from the references 

Thank you very much for your suggestions. The main focus of this paper is the expansion of the measurement range for borehole strainmeters. Currently, the strainmeters that have been used for range extension primarily achieve this by integrating mechanical adjustment structures at the sensor end, which is quite different from the electrical method of zero-setting in this paper, making direct comparisons unsuitable. Furthermore, most articles on component strainmeters analyze measurement performance based on solid tide data collected in the field, with very few conducting quantitative analyses in laboratory settings.

Therefore, the primary reference and comparison standard in this paper are the currently operational RZB-type strainmeters and the relevant industry standard documents. We believe that the electronic switch, as an extension module for traditional strainmeter devices, should ensure that the basic performance of the equipment is not compromised while enhancing functionality and applicability.

Based on the responses above and the descriptions in the text, we have demonstrated that the strainmeters equipped with electronic switches exhibits consistent measurement performance compared to the traditional version with mechanical switches in a laboratory environment, as for the overall measurement performance of the system, it is primarily related to the mechanical structure of the sensor and the host circuit. At the same time, it achieves automatic zero-setting and full-range measurement functions, fully utilizing the effective range of the sensor, and the testing linearity meets industry standards.

We hope you can recognize our conclusions. If you have any further questions, please feel free to contact us through the editorial office at any time!

Round 2

Reviewer 2 Report

Comments and Suggestions for Authors

The article has been corrected according to the requests.

However, supplementary files are in Chinese, so I cannot review them.